# Immune Checkpoint Inhibitor Use in Advanced Hepatocellular Carcinoma: A Real-World Analysis of Efficacy and Toxicity

**DOI:** 10.3390/cancers17183034

**Published:** 2025-09-17

**Authors:** Fode Tounkara, Deepak Sherpally, Khalid Mumtaz, Mina S. Makary, Russell F. Palm, Ashish Manne

**Affiliations:** 1Department of Biomedical Informatics, Center for Biostatistics, College of Medicine, The Ohio State University Wexner Medical Center, Columbus, OH 43210, USA; 2Department of Internal Medicine, New York Medical College, Metropolitan, New York, NY 10029, USA; dsherpally@nymc.edu; 3Division of Gastroenterology, Hepatology, and Nutrition, Department of Internal Medicine, The Ohio State University Wexner Medical Center, Columbus, OH 43210, USA; khalid.mumtaz@osumc.edu; 4Department of Radiology, The Ohio State University Wexner Medical Center, Columbus, OH 43210, USA; mina.makary@osumc.edu; 5Department of Radiation Oncology, The Ohio State University Comprehensive Cancer Center, Columbus, OH 43210, USA; russell.palm@osumc.edu; 6Division of Medical Oncology, Department of Internal Medicine, The Ohio State University Comprehensive Cancer Center, Columbus, OH 43210, USA; ashish.manne@osumc.edu

**Keywords:** HCC, liver cancer, immunotherapy, real-world, locoregional therapy, ALBI, Child–Pugh Score, immune-related adverse events, atezolizumab, immune-check point inhibitors

## Abstract

Medicines that boost the immune system, called immune checkpoint inhibitors (ICIs), are now used to treat liver cancer. Big studies often leave out people with more advanced liver problems, so we studied 53 patients treated at our hospital. On average, people lived about 7 months after starting ICIs, and some lived much longer, showing that responses can vary widely. Patients with healthier liver function and those who had earlier liver treatments did better overall. Side effects happened in about 1 in 5 patients but were usually manageable with standard care. These findings suggest that ICIs may be safe and effective even in patients who are not usually included in clinical trials, giving more people with liver cancer the chance to benefit from these newer, promising therapies and supporting their use in everyday practice.

## 1. Introduction

Hepatocellular carcinoma (HCC) represents the most prevalent form of primary liver malignancy, often associated with unfavorable prognoses, including a five-year survival rate of only 21% [1,2]. It is anticipated to become the third-leading cause of cancer-related mortality globally. The top etiologies for HCC per the Global Burden of Disease (GBD) database are hepatitis B (41%), hepatitis C (28.5%), alcoholic liver disease (18.4%), and metabolic-dysfunction-associated steatotic liver disease (MASLD) (6.8%) [3]. The predominant cause of HCC varies by region, with hepatitis B being most common in East Asia, hepatitis C in Japan, and both alcoholic and MASLD in the United States [3]. The global incidence of MASLD is rising, further contributing to the growing burden of HCC [4]. The disease typically presents at a median age of 65, with a peak incidence between 70 and 75 years, and occurs 2–3 times more frequently in males than females [3,5].

Multiple societies have proposed guidelines for appropriate treatment for HCC, which include the European Association for the Study of the Liver (EASL), the American Association for the Study of Liver Diseases (AASLD), the European Society for Medical Oncology (ESMO), and the National Comprehensive Cancer Network (NCCN) [6,7,8]. However, since 1999, the Barcelona Clinic Liver Cancer (BCLC) system, proposed by EASL, has been widely accepted as the cornerstone of HCC management worldwide [7,9]. It provides a detailed framework for treatment strategies based on disease stage, performance status, liver function based on Child–Turcotte–Pugh (CTP), and clinicopathological features, including the size and number of liver lesions, portal vein tumor thrombosis (PVTT), and distant metastasis. In the United States, the treatment approach is institution-specific, and it depends on the available resources such as transplant candidacy, thermal ablation, transarterial chemoembolization (TACE), transarterial radioembolization (TARE), and stereotactic body radiation therapy (SBRT) [6]. While locoregional therapy (LRT) practice patterns vary, with increasing utilization of TARE and SBRT in the US, the approach to systemic therapy is similar in most countries [10,11,12,13,14,15,16].

The systemic treatment landscape for HCC was dominated predominantly by sorafenib since the late 2000s, but now immune checkpoint inhibitors (ICI) have taken over with at least three combinations—atezolizumab and bevacizumab (IMBRAVE 150), the STRIDE regimen (single tremelimumab regular interval durvalumab, HIMALAYA), and ipilimumab and nivolumab (ChekMate 9DW) [17,18,19,20]. These pivotal trials enrolled patients with preserved liver function (mostly CTP-A), yet current regulatory approvals are agnostic to Child–Pugh status. In real-world experience, a substantial proportion of patients initiating systemic therapy have moderate liver dysfunction (CTP-B or worse), raising questions about generalizability and real-world effectiveness. Given these gaps, it is prudent to understand the outcomes of ICI-based therapies in real-world scenario, particularly across the CTP scores. This retrospective study aims to describe the real-world survival outcomes and safety profiles of patients with HCC treated with ICI-based therapies.

## 2. Methods

### 2.1. Patient Selection and Study Design

The research adhered to the Declaration of Helsinki and received approval from the Institutional Review Board (IRB) of the Ohio State University, Wexner Medical Center (protocol number 2021C0133, approved on 3 September 2021). The electronic medical records (EMR) from January 2017 to June 2023 were examined utilizing the International Statistical Classification of Diseases, 10th revision (ICD-10), diagnosis codes alongside the names of ICIs. All patients diagnosed with HCC who received at least one dose of ICI within the designated study timeframe were included. Data on demographics (age, race, and gender) and tumor-related clinicopathological information (number and size of primary lesions and presence of PVTT) were collected. Prior therapeutic interventions included systemic therapy (tyrosine-kinase inhibitors (TKIs)) and LRT (such as SBRT, TACE, TARE, and ablation), as well as laboratory data (complete blood counts including white blood cells (WBCs), red blood cells (RBCs), absolute neutrophilic count (ANC), absolute lymphocyte count (ALC), platelet count, hemoglobin, and biochemical parameters including total bilirubin (TBili), albumin, alkaline phosphatase (ALP), alanine transferase (ALT), and aspartate transferase (AST)).

### 2.2. The Study Outcomes

The primary outcome included the overall survival (OS), ICI-related overall survival (ICI-OS), the progression-free survival (PFS), and the incidence of immune-related adverse events (irAEs). The OS was defined as the time from cancer diagnosis to death or loss to follow-up, which ever came first; the ICI-OS was defined as the time from first ICI use to death or loss to follow-up and the PFS from the date of first ICI use to date of progression (patients who died were considered as progressed, and date of progression is their date of death). This distinction (OS vs. IS-ICI) was made to better understand the benefit of ICI with subsequent therapy for survival. We also reported changes in the blood counts—WBC, RBC, Hb, platelets, ANC, and ALC.

### 2.3. Statistical Analysis

Statistical analysis was conducted to identify associations between patient characteristics and survival outcomes, including PFS, OS, and OS-ICI. Descriptive statistics were employed to summarize the baseline characteristics of the study population. Patients’ characteristics are presented with counts and proportions for categorical variables and as mean ± SD and median (interquartile range) for quantitative variables.

To analyze survival data (OS, OS-ICI, and PFS), the Kaplan–Meier method was used to estimate survival curves for OS, ICI-OS, and PFS, and differences between groups were assessed using the log-rank test. To identify factors associated with survival outcomes, univariate Cox proportional hazards regression analyses were performed for each candidate variable. Variables with a *p*-value less than 0.10 in univariate analysis were included in multivariable Cox regression models, using a stepwise selection approach. In the final multivariable models, variables with a *p*-value less than 0.05 were considered statistically significant. Hazard ratios (HRs) and 95% confidence intervals (CIs) were reported for both univariate and multivariable analyses.

For secondary analyses, changes in hematologic parameters (such as white blood cell count, red blood cell count, hemoglobin, platelets, absolute neutrophil count, and absolute lymphocyte count) before and after ICI therapy were compared using paired or unpaired t-tests or non-parametric tests, as appropriate. The incidence of immune-related adverse events (irAEs) was summarized descriptively. Where feasible, logistic regression was used to explore associations between baseline characteristics and the occurrence of irAEs.

All statistical analyses were performed using version 4.5.0 of the R statistical computing environment from the R Foundation for Statistical Computing. All tests were two-sided, and a *p*-value less than 0.05 was considered statistically significant. No adjustments were made for multiple comparisons due to the exploratory nature of this study.

## 3. Results

### 3.1. Baseline Characteristics

Based on EMR search, 53 patients with HCC were included with a median age at a diagnosis of 66 years (range 40–86 years). Among them, 43 (81.1%) were males of a predominantly Caucasian (84.9%) race. Patients with multiple lesions (*n* > 3; 63.2%) and BCLC stage D (*n* = 36; 67.9%) dominated the cohort, with 41.5% having distant metastatic disease. Smoking history was significant (69.8%) in the study cohort, and approximately half (45.2%) had hepatitis C. Ascites was the most common (66%) cirrhotic decompensation noted, followed by hepatic encephalopathy (HE, 20.8%). Approximately half of the patients (*n* = 27; 50.9%) received first-line immunotherapy. Pembrolizumab, nivolumab, durvalumab, and atezolizumab were the different immunotherapeutic agents utilized in the patient population. TACE was the initial therapy used in 14 patients (26.4%), and the remaining patients received other LRT, including ablation (*n* = 6; 11.3%), TARE (*n* = 10; 18.8%), SBRT (*n* = 10; 18.8%), and surgery (*n* = 5; 9.4%). The rest of the baseline characteristics are outlined in Table 1.

### 3.2. Predictors of Progression-Free Survival

The median PFS was 4.6 months (CI: 3.2 to 7.83). Table 2 summarizes the univariate and multivariate analyses for key variables influencing the PFS. A full list of univariate analysis is provided in Appendix A. Several factors were significantly (*p* < 0.05) associated with inferior PFS including a history of alcohol use, a higher ALBI grade, receiving ICI beyond the first line, and an elevated neutrophil-to-lymphocyte ratio (NLR) and platelet-to-lymphocyte ratio (PLR). A trend toward statistical significance was also observed with prior LRT (HR: 1.74, *p* = 0.07).

Alcohol use (adjusted HR: 2.24; 95% CI: 1.20–4.17; *p* = 0.011) and the ALBI grade remained independent predictors of poor PFS. Patients with ALBI grade 2 or 3 had worse outcomes compared to grade 1, with adjusted HRs of 2.91 (95% CI: 1.12–7.52; *p* = 0.028) and 3.14 (95% CI: 1.12–8.80; *p* = 0.030), respectively. Receiving ICI beyond the first line showed a trend toward worse PFS (adjusted HR: 1.85; *p* = 0.062) but did not reach statistical significance.

To further illustrate the prognostic impact of the ALBI grade, Figure 1 presents Kaplan–Meier curves for PFS stratified by the ALBI grade. Grade 1 patients (median PFS = 24.8 months, 95% CI: 19.6 to not evaluable) had significantly better survival than grade 2 (17.4 months, 95% CI: 14.6 to 24.03) and grade 3 (11.8 months, 95% CI: 8.7 to 43.6).

We also examined changes in hematologic parameters to assess trends associated with progression. A statistically significant decrease in lymphocyte count (by 0.295, *p* = 0.014) was observed in patients with documented progression (*n* = 43) on ICI during the study period. The drop in RBC, Hb, and platelet counts and the rise in WBC count and ANC were insignificant (*p* > 0.05) (Appendix A).

### 3.3. Predictors of Overall Survival ICI

The median OS-ICI in our cohort was 7.4 months (CI: 4.87 to 11.3). Factors impacting OS-ICI on univariate analysis were a history of alcohol use, use of bevacizumab (bev) combination (with atezolizumab), and PLR (Table 3 and Appendix A). The age and size of the lesion (≤3 vs. >3 cm) had a trend towards significance. Three factors independently predicted OS-ICI, including a history of alcohol use, the bev combination, and the ALBI grade. A trend in OS-ICI was observed across ALBI grades without statistical significance (*p* = 0.2; Appendix A). Similarly, a trend consistent with the PFS analysis was observed, with median OS-ICI decreasing with worsening liver function: 15.1 months for ALBI grade 1, 7.6 months for grade 2, and 5.3 months for grade 3.

### 3.4. Predictors of Overall Survival

The median OS of the study cohort was 18.7 months (CI: 15.8 to 24.03). HCC patients with a history of LRT had a positive impact on OS on univariate analysis (Table 4). Importantly, two factors emerged as independent predictors of OS—prior LRT and ALBI grading. While a history of alcohol use showed a trend toward worse OS (adjusted HR: 1.68; *p* = 0.11), it did not reach statistical significance in the multivariate model. Moreover, though the ALBI grade did not reach statistical significance in the Kaplan–Meier analysis (*p* = 0.16; Appendix A), a consistent numerical trend was observed. There was also a trend of a median OS decrease with worsening liver function (median OS-ICI, ALBI grade 1 vs. grade 2 vs. grade 3 = 24.8 vs. 18.06 vs. 11.8 months, *p* = 0.16).

### 3.5. Predictors of irAE

We had ten (19%) patients with irAEs (three hepatitis, two pneumonitis, two dermatitis, one colitis, one myasthenia gravis, and one neutropenia). Most irAEs occurred in patients with CTP-A (*n* = 8; 80%). The median doses before irAE incidence were 2 (range: 1–28). Out of a total of six patients with severe irAEs (≥grade 3), two with hepatitis, two with pneumonitis, one with dermatitis, and one with myasthenia gravis had required hospitalization. ICI therapy was permanently discontinued in seven of these patients, while it was restarted in three. One patient (with pneumonitis) received dual immunotherapy (ipilimumab/nivolumab) after progressing on pembrolizumab. There were no irAE-related deaths.

Baseline (before first dose of ICI) clinicopathologic variables were not significantly different in the patients in the irAE group (*n* = 10) compared to the non-irAE group (*n* = 43) (Appendix A). Ascites were more common in the non-irAE group (72% vs. 40%) and showed a trend toward protective association with irAE development (*p* = 0.07). On univariate analysis, ascites had a protective effect against irAE occurrence (odds ratio: 0.26; 95% CI: 0.06–1.06; *p* = 0.06). However, no variables met the criteria for inclusion in a multivariate model (Appendix A). Evaluation of hematologic parameters revealed no statistically significant changes in blood counts at the time of irAE compared to baseline (Appendix A).

### 3.6. ALBI vs. CTP Score in HCC Management

A subgroup analysis was performed to study the association between ALBI grades and CTP categories (Table 5). Most patients (87%) with ALBI grade 1 were classified as CTP-A (88%), with only 13% as CTP-B and none as CTP-C. In contrast, patients with ALBI grade 3 have worse liver function, with only 14% categorized as CTP-A, while 50% and 36% were CTP-B and CTP-C, respectively. ALBI grade 2 represented a heterogeneous group, comprising 61% CTP-A, 35% CTP-B, and 3.2% CTP-C. These findings highlight a strong concordance between a worsening ALBI grade and higher CTP class, supporting the ALBI score as a robust objective marker of liver dysfunction.

## 4. Discussion

Immune-checkpoint-inhibitor-based combinations have transformed the treatment landscape for HCC, offering renewed hope in a disease historically marked by limited systemic options. Over the past five years, regimens such as atezolizumab–bevacizumab (IMbrave150), tremelimumab–durvalumab (HIMALAYA), and ipilimumab–nivolumab (CheckMate 9DW) have demonstrated clinical benefit in select patient populations. However, their applicability to broader, real-world cohorts remains uncertain. Our real-world study addresses this gap by identifying key predictors of treatment outcomes and highlighting significant deviations from the tightly controlled populations of pivotal trials. Notably, 45% of patients in our cohort had advanced liver disease reflected by CTP-B or C, rendering them ineligible for most landmark ICI trials, which primarily enrolled patients with well-compensated cirrhosis (CTP-A). This discrepancy underscores the urgent need for real-world evidence to consider treatment strategies for underrepresented populations with more advanced liver dysfunction. Our findings support the integration of real-world data to guide trial design, therapeutic selection, and the development of risk-adapted strategies—particularly for those with marginal hepatic reserve.

Among our key findings, the ALBI grade emerged as a robust and independent predictor of PFS, OS, and OS-ICI, outperforming the traditional Child–Pugh classification. While Child–Pugh remains widely used, our data show significant overlap and discordance between the two systems. For instance, 61% of patients with ALBI grade 2 and 14% with ALBI grade 3 were still classified as Child–Pugh-A, reflecting the limitations of Child–Pugh in capturing nuances of liver dysfunction. These findings reinforce the existing literature and support ALBI as a more objective, dynamic, and reproducible tool for clinical stratification and trial eligibility [21,22,23,24].

Alcohol-related liver disease was independently associated with worse PFS and OS-ICI, likely due to a combination of chronic hepatic injury, systemic inflammation, and immune dysregulation. This is consistent with prior observations and merits further investigation to determine whether alcohol-induced liver injury alters ICI pharmacodynamics or immune responsiveness. In contrast, prior LRT was significantly associated with improved overall survival. This finding may partly reflect favorable tumor biology and patient selection as patients eligible for LRT often present with lower tumor burden at recurrence or progression. However, it may also suggest a synergistic effect between LRT and ICI, potentially through enhanced tumor antigen presentation and immune priming from radiation-based treatments such as SBRT or TARE [25,26]. Importantly, these findings align with emerging preclinical evidence demonstrating that radiotherapy can augment the activity of checkpoint blockade by enhancing antigen release, dendritic cell activation, and T-cell priming [27,28,29,30,31,32]. Clinically, this raises the possibility that combining or sequencing LRT with ICI could provide additive or synergistic benefit as being investigated in multiple ongoing studies [25]. Additionally, the subset of patients treated with atezolizumab–bevacizumab experienced a clear OS-ICI benefit, consistent with results from IMbrave150 and reaffirming the efficacy of this combination, even in real-world settings.

The incidence of immune-related adverse events (irAEs) in our study was 19%, slightly lower than rates reported in clinical trials [17,18,19,20]. Most irAEs occurred in patients with preserved liver function (Child–Pugh-A), and no irAE-related mortality was observed. Interestingly, ascites appeared to be associated with a reduced risk of irAE development. However, this observation was based on very small patient numbers and should, therefore, be interpreted with caution. Rather than representing definitive associations, they are best viewed as hypothesis-generating signals that warrant further study in larger, prospective cohorts. Future validation could help clarify whether ascites or other irAE patterns have a mechanistic or prognostic role in this setting. Overall, irAEs were manageable, further supporting the feasibility of ICI use in everyday practice with appropriate monitoring.

This study has several limitations. While our findings offer initial insights into the associations between clinical characteristics and survival outcomes, we acknowledge that the study’s small sample size (53 patients from a single institution) limits the statistical power and generalizability of the results. As such, these observations should be considered exploratory and hypothesis-generating rather than definitive. Heterogeneity in treatment regimens and the timing of ICI use (especially pre- and post-approval of key trials) may introduce bias. We did not have enough patients to compare the performance of current available treatment combinations. While we attempted to control for key covariates, unmeasured confounders could influence outcomes. Lastly, the observational nature of the study precludes causal inference. Despite its limitations, this study provides valuable real-world insights into the efficacy and safety of ICI-based therapies across a more diverse HCC population than typically represented in clinical trials.

## 5. Conclusions

Our findings support the use of ICI-based therapies in HCC, particularly when guided by objective liver function assessments such as the ALBI grade. These data suggest that patients with ALBI grade 1 derive the most benefit from immunotherapy, while caution is warranted in ALBI grades 2–3, where outcomes are less favorable. Bevacizumab combination and prior locoregional therapies may enhance response, and alcohol-related liver disease remains a critical modifier of prognosis. Future prospective studies are needed to validate these findings in larger, multicenter cohorts and to explore biomarker-guided strategies that integrate liver function metrics with immune profiling. Stratifying patients not only by tumor burden but also by hepatic reserve and inflammatory status will be essential to optimize immunotherapy outcomes in HCC.

## Figures and Tables

**Figure 1 cancers-17-03034-f001:**
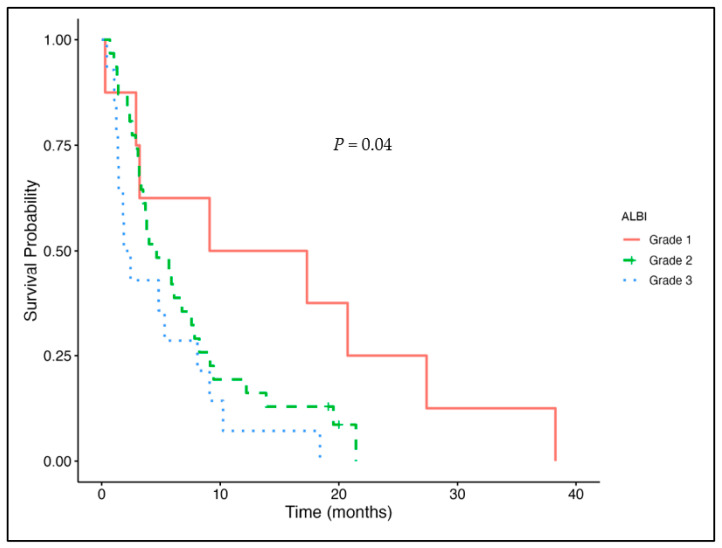
Kaplan–Meier curve for progression-free survival by ALBI grade.

**Table 1 cancers-17-03034-t001:** Baseline characteristics of the study population (*n* = 53).

Characteristic	N (%)
Age at diagnosis	66 years (40–86)
Gender (male)	81.1%
Race	Caucasian: 84.9%, African-American: 13.2%, other: 1.8%
Hepatitis C status	Positive: 45.2%, negative: 54.8%
Smoking status	69.8%
Barcelona Clinic Liver Cancer Staging Category	D: 67.9%, C: 22.6%, B: 9.4%
Lines of immune-checkpoint inhibitors used	1: 50.9%, 2: 18.8%, 3: 16.9%, 4: 5.6%, >4: 7.5%
History of systemic therapy	35.8%
Immune-checkpoint inhibitor agent used	Atezolizumab—32%, Durvalumab—2%, Nivolumab—62%, Pembrolizumab—3.7%
Bevacizumab combination *	24.5%
Alpha-fetoprotein at the use of immune-checkpoint inhibitors	404 (1.4–1,000,000)
Number of lesions (≤3)	35.8%
Portal vein tumor thrombosis	Negative: 54.7%
Distant metastatic disease	41.5%
Locoregional therapy used	Ablation: 11%, TACE: 26.4%, TARE: 19%, SBRT: 19%, Surgery: 9%
Previous locoregional therapy	49%
Ascites ^#^	66%
Encephalopathy ^#^	Negative: 79.2%
Child–Pugh Score ^#^	A—53%, B—36%, C—11%
White blood cell count ^#m^	5.54 (1.97–14.76) K/uL
Red blood cell count ^#m^	4 (2.5–6.2) M/uL
Hemoglobin ^#m^	12.4 (7–16.3) g/dL
Platelet count ^#m^	163 (43–581) K/uL
Absolute neutrophil count ^#m^	3.63 (1.1–12.1) K/uL
Lymphocyte count ^#m^	1.1 (0.16–4.3)K/uL
Platelet lymphocyte ratio ^#m^	163 (25.1–488.2)
Neutrophil lymphocyte ratio ^#m^	3.54 (0.72–26.06)
Total bilirubin ^#m^	1.1 (0.3–5.3) mg/dL
Albumin ^#m^	3.3 (2.3–4.7) g/dL
Albumin–bilirubin (ALBI) score ^#^	1: 20%, 2: 63%, 3: 17%
Immune-related adverse event	Hepatitis: 3/10, pneumonitis: 2/10, dermatitis: 2/10
Immune-related adverse event required hospitalization	6/10
Immune-checkpoint inhibitor discontinuation reason	Disease progression: 66%, Immune-related adverse event: 13.2%, Death: 16.9%
Next line of therapy ^##^	TKIs—21%, TACE—4%, TARE-2%, capecitabine—6%, DI-2%, supportive care or death—65%
First immune-checkpoint inhibitor dose after 2020	55%

* with atezolizumab, # before immune-checkpoint inhibitors, m—median level, ## in progressed group, TACE—transarterial chemoembolization, TARE—transarterial radioembolization (TARE), SBRT—stereotactic brachy radiotherapy, TKI—tyrosine kinase inhibitor, and DI—dual immunotherapy.

**Table 2 cancers-17-03034-t002:** Predictors of progression-free survival (PFS) on univariate and multivariate analysis.

Factor	Univariable (PFS)			Multivariable (PFS)		
	HR	95% CI	*p*-Value	Adjusted HR	95% CI	*p*-Value
**Alcohol History**						
No	-	-		-	-	
Yes	2.10	1.15, 3.83	**0.016**	2.24	1.20, 4.17	**0.011**
**Line of ICI**						
1	-	-		-	-	
>1	1.74	0.96, 3.17	0.070	1.85	0.97, 3.53	0.062
**Previous Locoregional Therapy**						
no	-	-				
yes	1.74	0.96, 3.17	0.070			
**ALBI Grade**						
Grade 1	-	-		-	-	
Grade 2	2.21	0.89, 5.52	0.089	2.91	1.12, 7.52	**0.028**
Grade 3	3.51	1.28, 9.63	**0.015**	3.14	1.12, 8.80	**0.030**
PLR *	1.00	1.00, 1.01	**0.045**			
NLR *	1.08	1.01, 1.15	**0.033**			

HR = hazard ratio; CI = confidence interval. Bold—significant *p* value. * These factors did not meet the criteria to do multivariate analysis. ALBI—Albumin-Bilirubin, PLR—platelet-to-lymphocyte ratio; NLR—neutrophil-to-lymphocyte ratio.

**Table 3 cancers-17-03034-t003:** Predictors of immune-checkpoint-inhibitor-specific overall survival (OS-ICI) on univariate and multivariate analysis.

Factor	Univariable (OS-ICI)			Multivariable (OS-ICI)		
	HR	95% CI	*p*-Value	Adjusted HR	95% CI	*p*-Value
**Age**	0.97	0.94, 1.00	0.060	0.96	0.93, 1.01	0.086
**Alcohol History**						
No	-	-		-	-	
Yes	2.07	1.11, 3.84	**0.021**	2.13	1.09, 4.19	**0.028**
**Bevacizumab Combination**						
No	-	-		-	-	
Yes	0.46	0.23, 0.93	**0.032**	0.37	0.16, 0.86	**0.021**
**Size of the Primary Lesion (in cm)**						
≤3	-	-		-	-	
>3	0.50	0.25, 1.00	0.051	0.53	0.25, 1.13	0.10
**ALBI Grade**						
Grade 1	-	-		-	-	
Grade 2	1.69	0.72, 3.97	0.2	2.74	1.10, 6.80	**0.030**
Grade 3	2.35	0.91, 6.03	0.076	6.06	1.99, 18.5	**0.002**
PLR *****	1.00	1.00, 1.01	**0.049**			

* These factors did not meet criteria to do multivariate analysis. PLR—platelet-to-lymphocyte ratio. HR- hazard ratio, CI—confidence interval, ALBI—Albumin-Bilirubin, Bold—significant *p* value.

**Table 4 cancers-17-03034-t004:** Predictors of overall survival (OS) on univariate and multivariate analysis.

Factor	Univariable (OS)			Multivariable (OS)		
	HR	95% CI	*p*-Value	Adjusted HR	95% CI	*p*-Value
**Alcohol History**						
No	-	-	-	-	-	-
Yes	1.74	0.94, 3.22	0.077	1.68	0.89, 3.15	0.11
**Previous Locoregional Therapy**						
None	-	-	-	-	-	-
Yes	0.53	0.29, 0.95	**0.033**	0.43	0.22, 0.83	**0.012**
**ALBI Grade**						
Grade 1	-	-	-	-	-	-
Grade 2	2.33	0.96, 5.64	0.061	2.63	1.07, 6.46	**0.035**
Grade 3	2.05	0.80, 5.20	0.13	2.84	1.03, 7.87	**0.044**

HR—hazard ratio, CI—confidence interval, ALBI—Albumin-Bilirubin, Bold—significant *p* value.

**Table 5 cancers-17-03034-t005:** Association between Albumin-Bilirubin grades and Child–Pugh Category groups.

Child–Pugh Category	ALBI Grades			
	1 (*n* = 8)	2 (*n* = 31)	3 (*n* = 14)	*p*-Value
A	7 (88%)	19 (61%)	2 (14%)	0.01
B	1 (12%)	11 (31%)	7 (50%)	
C	0	1 (2%)	5 (36%)	

## Data Availability

For ethical reasons, the data presented in this study are available upon request from the corresponding author.

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
