# Peer review of "Immune Checkpoint Inhibitor Use in Advanced Hepatocellular Carcinoma: A Real-World Analysis of Efficacy and Toxicity"

_cancers, 2025, doi:10.3390/cancers17183034_

Round 1
Reviewer 1 Report
Comments and Suggestions for Authors
The authors present a retrospective analysis of 53 HCC patients treated with immune checkpoint inhibitors (ICIs) at a single center. The reported median PFS, OS-ICI, and OS were 18.7 months, 7.4 months, and 4.6 months, respectively. Higher ALBI grade and a history of alcohol use were independently associated with worse PFS and OS-ICI, while prior locoregional therapy (LRT) significantly improved OS.
The study is of interest but is limited by its small sample size (53 patients from a single hospital), resulting in a very low level of evidence.
The authors conclude that this study provides real-world insights into the efficacy and safety of ICI-based therapies in a diverse HCC population. However, such a conclusion cannot be drawn without comparison to other therapies. In the introduction, the authors state: “This retrospective study aims to evaluate the utility, effectiveness, and safety of ICIs in a real-world HCC population.” Yet, no true measure of effectiveness is presented. What is considered “effectiveness” in this study—an OS of 4.9 months? Is that outcome favorable or poor, and how would it compare to an alternative therapy?
The only relevant finding appears to be the association of certain variables with survival.
Moreover, the survival data are inconsistent. How can PFS (18.7 months) be longer than OS (4.6 months)? If patients lived a median of only 4.6 months, how could their median progression-free survival be 18.7 months? Did the authors censor deaths not directly attributed to HCC? This point requires clarification.
Finally, Table 1 should be revised. All terms should be written out in full (without abbreviations), and measurement units must be provided for all laboratory values.
Author Response
1. The study is of interest but is limited by its small sample size (53 patients from a single hospital), resulting in a very low level of evidence.
Response:
Thanks for the comment. We understand that it is a low number, which in itself is a significant story here. In the United States, we are not utilizing systemic therapy sufficiently and are relying too heavily on local therapies. Our real-world practice bears little resemblance to trial patient populations. However, we addressed this issue in our discussion in lines 297 to 301. It now reads, While our findings offer initial insights into the associations between clinical characteristics and survival outcomes, we acknowledge that the study’s small sample size (53 patients from a single institution) limits the statistical power and generalizability of the results. As such, these observations should be considered exploratory and hypothesis-generating rather than definitive.
2.
A) The authors conclude that this study provides real-world insights into the efficacy and safety of ICI-based therapies in a diverse HCC population. However, such a conclusion cannot be drawn without comparison to other therapies. In the introduction, the authors state: “This retrospective study aims to evaluate the utility, effectiveness, and safety of ICIs in a real-world HCC population.”Yet, no true measure of effectiveness is presented. What is considered “effectiveness” in this study—an OS of 4.9 months? Is that outcome favorable or poor, and how would it compare to an alternative therapy?
The only relevant finding appears to be the association of certain variables with survival.
B) Moreover, the survival data are inconsistent. How can PFS (18.7 months) be longer than OS (4.6 months)? If patients lived a median of only 4.6 months, how could their median progression-free survival be 18.7 months? Did the authors censor deaths not directly attributed to HCC? This point requires clarification.
Response:
Thanks for the constructive criticism. As these two comments were related, we answered them together.
We sincerely apologize for the confusion. There was a typo in the abstract, which is unfortunate. The order of the survivals was wrong. The order should have been PFS, OS-ICI, and PFS instead of OS, OS-ICI, and PFS. We corrected it now. The text of the manuscript was accurate.
Regarding comparing therapies. The HIMALAYA trial and approval of the durvalumab and tremelimumab were towards the end of the study period, and there were not enough patients on it to compare it with atezolizumab and bevacizumab. We added a line addressing it in ‘Discussion’ section, in lines 303-304 - We did not have enough patients to compare the performance of current available treatment combinations.
Finally, regarding our line in the ‘Introduction’ section, we replaced This retrospective study aims to evaluate the utility, effectiveness, and safety of ICIs in a real-world HCC population with This retrospective study aims to describe the real-world survival outcomes and safety profiles of patients with HCC treated with ICI-based therapies in lines 90-91.
3. Finally, Table 1 should be revised. All terms should be written out in full (without abbreviations), and measurement units must be provided for all laboratory values.
Response:
Thanks for the comment. We expanded most of the abbreviations. We are happy to some of them we had to use in the legends back to the table if it is not acceptable. We added the units for lab values as requested.
Reviewer 2 Report
Comments and Suggestions for Authors
There are several comments to be addressed.
1) In the abstract, the median PFS, OS-ICI, and OS were 18.7 months(m), 7.4 m, and 4.6 m, respectively. It does not make sense.
2) Multivariable models may be overfitted; wide CIs highlight underpowered analyses.
3) irAE observations (including ascites “protective” effect) are interesting but limited by very small numbers; should be framed as hypothesis-generating.
4) Association of prior locoregional therapy with improved survival may reflect selection bias, because eligibility for locoregional therapy might reflect the lesser tumor burden.
5) Along with comment 4, immune-priming should be also discussed.
Author Response
1) In the abstract, the median PFS, OS-ICI, and OS were 18.7 months(m), 7.4 m, and 4.6 m, respectively. It does not make sense.
- Response:
- Thanks for the comment. There was a typo in the abstract, which is unfortunate. The order of the survivals was wrong. The order should have been PFS, OS-ICI, and PFS instead of OS, OS-ICI, and PFS. We corrected it now. The text of the manuscript was accurate.
2) Multivariable models may be overfitted; wide CIs highlight underpowered analyses.
- Response:
- We agree with the reviewer that the multivariable models are limited by the small sample size. We addressed it in our discussion part (lines 297 to 301) While our findings offer initial insights into the associations between clinical characteristics and survival outcomes, we acknowledge that the study’s small sample size (53 patients from a single institution) limits the statistical power and generalizability of the results. As such, these observations should be considered exploratory and hypothesis-generating rather than definitive.
3) irAE observations (including ascites “protective” effect) are interesting but limited by very small numbers; should be framed as hypothesis-generating.
- Response:
- We appreciate the comment. We addressed it in our discussion part (lines 287-292). Now it reads, Interestingly, ascites appeared to be associated with a reduced risk of irAE development. However, this observation was based on very small patient numbers and should therefore be interpreted with caution. Rather than representing definitive associations, they are best viewed as hypothesis-generating signals that warrant further study in larger, prospective cohorts. Future validation could help clarify whether ascites or other irAE patterns have a mechanistic or prognostic role in this setting.
4) Association of prior locoregional therapy with improved survival may reflect selection bias, because eligibility for locoregional therapy might reflect the lesser tumor burden.
5) Along with comment 4, immune-priming should be also discussed.
- Response:
- Thank you for the comment. We tried to address this in the discussion section earlier but we expanded and revised it as follows in lines 273-282, This finding may partly reflect favorable tumor biology and patient selection, as patients eligible for LRT often present with lower tumor burden at recurrence or progression. However, it may also suggest a synergistic effect between LRT and ICI, potentially through enhanced tumor antigen presentation and immune priming from radiation-based treat-ments such as SBRT or TARE [25,26]. Importantly, these findings align with emerging preclinical evidence demonstrating that radiotherapy can augment the activity of check-point blockade by enhancing antigen release, dendritic cell activation, and T-cell priming [27-32]. Clinically, this raises the possibility that combining or sequencing LRT with ICI could provide additive or synergistic benefit as being investigated in multiple on-going studies [25].
Reviewer 3 Report
Comments and Suggestions for Authors
This study is important, well done and important. Your conclusion is that future prospective studies are needed. Are you currently collaborating with other liver centers to prospectively (your paper was retrospective) substantiate your findings.
Regardless it is abundantly clear that ingestion of alcohol, even in relatively small amounts, may give rise to a spectrum of malignancies, in particular HCC.
Author Response
This study is important, well done and important. Your conclusion is that future prospective studies are needed. Are you currently collaborating with other liver centers to prospectively (your paper was retrospective) substantiate your findings.
- Response:
- Thanks for the comments. Yes. We started a prospective protocol to collect data on patients receiving systemic therapy and the plan is to have international collaborations.
Round 2
Reviewer 1 Report
Comments and Suggestions for Authors
-
Reviewer 2 Report
Comments and Suggestions for Authors
Authors addressed raised issues appropriately.